# Towards Generic Simulation for Demanding Stochastic Processes

**Demetris Koutsoyiannis *** and **Panayiotis Dimitriadis**

Department of Water Resources and Environmental Engineering, School of Civil Engineering, National Technical University of Athens, 15780 Athens, Greece; pandim@itia.ntua.gr
* Correspondence: dk@itia.ntua.gr

**Abstract:** We outline and test a new methodology for genuine simulation of stochastic processes with any dependence structure and any marginal distribution. We reproduce time dependence with a generalized, time symmetric or asymmetric, moving-average scheme. This implements linear filtering of non-Gaussian white noise, with the weights of the filter determined by analytical equations, in terms of the autocovariance of the process. We approximate the marginal distribution of the process, irrespective of its type, using a number of its cumulants, which in turn determine the cumulants of white noise, in a manner that can readily support the generation of random numbers from that approximation, so that it be applicable for stochastic simulation. The simulation method is genuine as it uses the process of interest directly, without any transformation (e.g., normalization). We illustrate the method in a number of synthetic and real-world applications, with either persistence or antipersistence, and with non-Gaussian marginal distributions that are bounded, thus making the problem more demanding. These include distributions bounded from both sides, such as uniform, and bounded from below, such as exponential and Pareto, possibly having a discontinuity at the origin (intermittence). All examples studied show the satisfactory performance of the method.

**Keywords:** stochastics; stochastic processes; stochastic simulation; Monte Carlo simulation; long range dependence; persistence; Hurst–Kolmogorov dynamics; climacogram; cumulants; intermittence

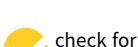



## 1. Introduction

Reviews on the historical evolution of simulation of stochastic processes, with its different schools, have recently been provided by Koutsoyiannis [1,2] and Beven [3]. In most scientific disciplines, the dominant methods are those of the so-called time series school, which developed families of models, known by the acronym ARMA (autoregressive–moving average). These are also called Box–Jenkins models, after the influential book by these authors [4], thus confirming Stigler's law of eponymy [5], because, in fact, they were introduced earlier by Whittle [6–8]. Despite their popularity, these models have several problems, such as their lack of parsimony (except for the simplest of them, e.g., the ARMA(1,1), summarized in the Appendix A), as well as the inability to model long-range dependence (LRD) and to simulate non-Gaussian processes. On the other hand, both of these features are profoundly present in most geophysical processes [9]. An extension of these models, applicable to processes with LRD, was proposed by Hosking [10] under the acronym ARFIMA (with the letter 'F' standing for fractional differencing and the letter 'I' for integrated). Again, these are good for Gaussian processes. Koutsoyiannis (2000) [11] introduced the symmetric moving average (SMA) scheme to replace ARMA models with a generic approach (more recently advanced in [12]), capable of reproducing any aspect of time dependence, short-range (SRD) or long-range (LRD), in a parsimonious manner, i.e., with a low number of parameters that are estimated from the data. This scheme can also preserve the skewness of non-Gaussian processes, but has difficulty in dealing with

higher-order moments, particularly with strongly intermittent processes, such as rainfall at small time scales.

For the latter, point process (clustered) models were devised [13–16]. One advantage of these types of models is the mechanistic representation of certain aspects of the process, such as the arrival and cease of a storm event. The disadvantages are mainly focused on the preservation of the dependence structure at multiple scales and their difficulty in application in multivariate or multiscale schemes. For this reason, Koutsoyiannis et al. [17], even though they used a 3D extension of a point process model (the so-called Gaussian displacement spatial–temporal rainfall [18]), resorted to a linear generation scheme for an application to multivariate rainfall disaggregation.

Several other modelling schemes use transformations of the process of interest, mostly within a copula context [19–22], with the most widely applied transformation resulting in a Gaussian process (normalization) [23,24]. However, such transformation schemes inherit some of the limitations of the parent process. For example, it is well known that a Gaussian process is necessarily symmetric in time and, thus, cannot capture time directionality, otherwise known as irreversibility or time's arrow [25]. On the other hand, it is known that, in several natural processes, time's arrow is present [26,27], and to reproduce it, we need processes with asymmetric distributions, which can also exhibit asymmetry in time.

A more general algorithm for generation of any type of marginal distribution was recently proposed by Lombardo et al. [28], but only under the condition of Markov dependence, thus leaving out problems with more complex dependence, including LRD. Recent advances include the use of machine learning methods in stochastic simulation, e.g., [29], which, however, have the disadvantages of being implicit in their mathematical structure, and non-parsimonious.

For these reasons, it is necessary to develop *genuine stochastic simulation* procedures, which will be able to generate non-Gaussian processes without any transformation to a Gaussian or other distribution. Such procedures have already been discussed in earlier works, referring to the explicit preservation of four moments in a time-symmetric setting [30] as well as preservation of distributions in terms of cumulants, rather than moments [2,31]. However, the general idea of the latter works has never been applied in practice to test its effectiveness. This is the subject of this paper.

The new methodology advances the state-of-the-art in stochastic generation by providing a general framework, capable of dealing with challenging Monte Carlo applications within geophysics, engineering, and other fields. The merits of the methodology rely on its ability to cope with the following aspects:

1. Complex dependence structures that extend way beyond the Markov dependence, and incorporate long-range dependence and short-scale fractal (smoothness/roughness) behavior. This is achieved by using a symmetric moving average scheme, which can involve a large number of white noise terms, with their weights determined in an explicit analytical manner.
2. Marginal distributions that extend beyond Gaussian and incorporate heavy tails, boundedness, and intermittence. This is achieved by using an appropriate number of cumulants, analytically determined from the distribution function, thus resulting in genuine simulation of the process (without a transformation).
3. Time asymmetry (irreversibility), achieved by using a non-Gaussian distribution function, combined with an asymmetric moving average scheme, with the weights again determined in an explicit analytical manner.

In the following sections, we outline the new methodology for genuine simulation (Section 2), and illustrate it in a number of synthetic and real-world applications (Section 3). In addition, we study the problem of approximating any distribution, if a number of its cumulants are known, in a manner that can readily support the generation of random numbers from that approximation (Section 2.5 and discussion in Section 4). Such approximation is suitable for analytical derivations, as well as for stochastic simulation in geophysical and engineering applications and beyond.

The simulation model developed is a linear stochastic model. As nonlinearity is fashionable, some may think that the linearity of the approach proposed is a limitation or even a severe drawback. The reality however is different because linearity and nonlinearity have different meaning in deterministic and stochastic approaches. In the latter, linearity is a powerful characteristic, enabling its extension in demanding problems, such as multivariate models and coupling of models of different temporal or spatial scales [32] (also known as downscaling or disaggregation). In this respect, it is relevant to recall the notion of Wold decomposition of stochastic processes. Specifically, Wold [33,34] proved that any stochastic process (even though he referred to it as a time series) can be decomposed into a regular process (i.e., a process linearly equivalent to a white noise process) and a predictable process (i.e., a process that can be expressed in terms of its past values). Thus, nonlinearity is relevant to the predictable part, as this is purely deterministic, while for the regular part linearity suffices.

## 2. Methods

### 2.1. Preliminaries

We denote $\underline{x}$ a stochastic (random) variable (underlining its symbol in order to distinguish it from a regular variable), $F(x) := P\{\underline{x} \leq x\}$ its probability distribution function, $\overline{F}(x) := 1 - F(x) = P\{\underline{x} > x\}$ its tail function (probability of exceedance) and $f(x) := \mathrm{d}f(x)/\mathrm{d}x$ its density function. Furthermore, we denote $\underline{x}(t)$ a stochastic process at continuous time $t$ (i.e., a family of stochastic variables $\underline{x}$ indexed by time $t$) and $\underline{x}_\tau := \frac{1}{D} \int_{(\tau-1)D}^{\tau D} \underline{x}(t)\mathrm{d}t$ its discrete time representation at equidistant times with temporal resolution $D$, i.e., $t_\tau = \tau D$, for an integer $\tau$. In a discrete-time stochastic process, it is convenient to define the *return period*, $T$, of the event $\{\underline{x}_\tau > x\}$ as the average time between two occurrences of the event. It is shown [2] that the following relationship holds true for any stochastic process (irrespective of time dependence):

$$\frac{T(x)}{D} = \frac{1}{\overline{F}(x)} \tag{1}$$

In other words, this one-to-one correspondence allows the return period to be used in place of the tail function or the distribution function in several applications (e.g., in probability plots); this has been the case for many years, particularly in engineering applications.

### 2.2. Moments and Cumulants

The expectation of any function $g(\underline{x})$ of the stochastic variable $\underline{x}$ is defined as:

$$\mathrm{E}[g(\underline{x})] := \int_{-\infty}^{\infty} g(x)f(x)\mathrm{d}x \tag{2}$$

where we remind that $g(\underline{x})$ is a stochastic variable per se. For $g(\underline{x}) = \underline{x}^p$, we get the *non-central moment* of order $p$ (or p*th raw moment* or p*th moment about the origin*):

$$\mu'_p := \mathrm{E}[\underline{x}^p] \tag{3}$$

with the particular case $p = 1$ defining the mean:

$$\mu := \mu'_1 = \mathrm{E}[\underline{x}] \tag{4}$$

The central moment of order $p$ is the expectation of $g(\underline{x}) = (\underline{x} - \mu)^p$:

$$\mu_p := \mathrm{E}[(\underline{x} - \mu)^p] \tag{5}$$

with the particular case $p = 2$ defining the variance:

$$\mu_2 \equiv \gamma := \mathrm{E}\left[(\underline{x} - \mu)^2\right] =: \sigma^2 \tag{6}$$

where its square root $\sigma$ is the standard deviation.

By choosing $g(\underline{x}) = \mathrm{e}^{t\underline{x}}$ for any $t$, the logarithm of the resulting expectation is called the *cumulant generating function*:

$$K(t) := \ln \mathrm{E}\left[e^{t\underline{x}}\right] \tag{7}$$

The power series expansion of the cumulant generating function, i.e.,

$$K(t) = \sum_{p=1}^{\infty} \kappa_p \frac{t^p}{p!} \tag{8}$$

defines the *cumulants* $\kappa_p$. It is noted that the cumulants were introduced by Thielle as early as in 1889 [35] and refined in 1899 [36,37] under the name *half-invariants*. The name *cumulants* was first used by Fisher [38] by the suggestion of Hotelling [39].

Cumulants are related to non-central moments of the same and lower order by:

$$\mu'_p = \sum_{i=0}^{p-1} \binom{p-1}{i} \kappa_{p-i} \mu'_i, \quad \kappa_p = \mu'_p - \sum_{i=1}^{p-1} \binom{p-1}{i} \kappa_{p-i} \mu'_i \tag{9}$$

with $\mu'_0 = 1$. A simple proof of these equations has been provided by Smith (1995) [40], but the recursive relationships had been already implied by Thielle [35,37]. Note that Equation (9) links cumulants with non-central moments. The relationship of cumulants with central moments is generally more complex, but for small $p$ it takes the following simple forms:

$$\kappa_0 = \mu_1 = 0, \quad \kappa_1 = \mu'_1 = \mu, \quad \kappa_2 = \mu_2, \quad \kappa_3 = \mu_3, \quad \kappa_4 = \mu_4 - 3\mu_2^2 \tag{10}$$

Equation (9) is very powerful as it allows simple calculation of cumulants from non-central moments and vice versa in a recursive manner. Notably, for the calculation of the moment or the cumulant of order $p$, the sums appearing in Equation (9) contain terms of order not higher than $p$.

The importance of cumulants results from their homogeneity and additivity properties, as seen in Table 1. Most importantly, for a stochastic variable that is the linear combination (weighted sum) of $r$ independent variables, the cumulants of the resultant are also a linear combination of the cumulants of the constituents. On the other hand, application of conditioning, also contained in Table 1, is similarly useful as it allows simulation of distributions that are mixtures of other distributions or have discontinuities in their distribution functions. As seen in Table 1, the effect of conditioning is more easily expressed in terms of moments, but Equation (9) readily allows the subsequent evaluation of cumulants.

All common distribution functions used in a wide range of stochastic applications have elegant analytical expressions either of their moments or the cumulants of any order, and in some cases of both. These are gathered in Table 2 for distributions with finite domain, in Table 3 for distributions with infinite domain, but with all their moments finite, and in Table 4 for the heavy-tailed distributions with upper-tail index $\xi$; in the latter case, both moments and cumulants exist for $p < 1/\xi$ and are infinite for larger $p$. The following notes apply to these tables:

1.  The meaning of the parameters is the following.

    (a) Dimensional parameters, with dimensions identical to those of the stochastic variable $\underline{x}$: $\mu$: mean; $\sigma > 0$: standard deviation; $\lambda > 0$: scale parameter; $a, b$: lower and upper bound of $\underline{x}$.
    (b) Dimensionless parameters: $\xi > 0$: upper-tail index; $\zeta > 0$: lower-tail index; $\varsigma > 0$: additional shape parameter, $P_i \in [0, 1]$: probability.

2. The meaning of constants and standard functions is this: $\gamma$: Euler constant; $B_p$: Bernoulli number of order $p$; $\delta(x)$: Dirac delta function of $x$; $\Gamma(a)$: gamma function of $a$; $\psi(a)$: digamma function of $a$; $B(a, b)$: beta function of $a, b$.

3. Distributions named "half" have their "full" version whose density $f(x)$ and tail function $\overline{F}(x)$ are obtained by dividing those given in the tables by 2. The "half" version given in the tables corresponds to $\underline{x} \geq 0$, while in the "full" version $\underline{x} \in \mathbb{R}$. The moments $\mu'_p$ of the "full" version is: (a) for even $p$, 0; (b) for odd $p$, equal to those of half version.

4. All other distributions, defined for $\underline{x} \geq 0$ but not named "half", can also be extended to the whole real line by replacing $x$ with $|x|$ and dividing $f(x)$ by 2. Again, the moments $\mu'_p$ of this extended version is: (a) for even $p$, 0; (b) for odd $p$, equal to those of original version.

**Table 1.** Typical operations useful in simulation and their mathematical handling.

| Operation | Mathematical Relationship | Eqn. no. |
|---|---|---|
| Shift of origin | $\kappa_p[\underline{x} + c] = \begin{cases} \kappa_1[\underline{x}] + c & p = 1 \\ \kappa_p[\underline{x}] & p > 1 \end{cases}$ | (11) |
| Multiplication by a constant $(a)$ | $\kappa_p[a\underline{x}] = a^p \kappa_p[\underline{x}]$ | (12) |
| Linear combination of independent variables | $\kappa_p[a_1\underline{x}_1 + \ldots + a_r\underline{x}_r] = a_1^p \kappa_p[\underline{x}_1] + \ldots + a_r^p \kappa_p[\underline{x}_r]$ | (13) |
| Conditioning on an event $A_1$ with probability $P_1 := P(A_1)$, where the complementary event $A_2$ has probability $1 - P_1 = P(A_2)$ | $\mu'_p[\underline{x}] = P_1\mu'_p[\underline{x}|A_1] + (1 - P_1)\mu'_p[\underline{x}|A_2]$ | (14) |
| Conditioning on an event $A_1$ with probability $P_1 := P(A_1)$, where $\underline{x} = c$ (constant) upon the complementary event $A_2$ | $\mu'_p[\underline{x}] = P_1\mu'_p[\underline{x}|A_1] + (1 - P_1)c^p$ | (15) |
| Conditioning on an event $A_1$ with probability $P_1 := P(A_1)$, where $\underline{x} = 0$ upon the complementary event $A_2$ | $\mu'_p[\underline{x}] = P_1\mu'_p[\underline{x}|A_1]$ | (16) |

**Table 2.** Non-central moments and cumulants of common distributions with finite domain (all moments and cumulants exist).

| Name, Domain | Probability Density or Distribution Function | Moments, $\mu'_p$ | Cumulants, $\kappa_p$ |
|---|---|---|---|
| Impulse, $\underline{x} = \mu$ | $f(x) = \delta(x - \mu)$ | $\mu^p$ | $\begin{cases} \mu & p = 1 \\ 0 & p > 1 \end{cases}$ |
| Finite number of impulses, $\underline{x} \in \{x_1, \ldots, x_n\}$ | $f(x) = \sum_{i=1}^{n} P_i\delta(x - x_i)$ | $\sum_{i=1}^{n} P_i x_i^p$ | |
| Uniform, $a \leq \underline{x} \leq b$ | $f(x) = \frac{1}{b-a}$ | $\frac{b^{p+1} - a^{p+1}}{(p+1)(b-a)}$ | $\begin{cases} \mu'_1 = \frac{a+b}{2} & p = 1 \\ \frac{(b-a)^p B_p}{p} & p \text{ odd} \\ 0 & p \text{ even} \end{cases}$ |
| Beta, $0 \leq \underline{x} \leq b$ | $f(x) = \frac{\left(\frac{x}{b}\right)^{\zeta-1}\left(1-\frac{x}{b}\right)^{\varsigma-1}}{B(\zeta,\varsigma)}$ | $\frac{\Gamma(\zeta+\varsigma)\,\Gamma(p+\zeta)}{\Gamma(\zeta)\Gamma(p+\zeta+\varsigma)}b^p$ | |
| Kumaraswamy, $0 \leq \underline{x} \leq b$ | $F(x) = 1 - \left(1 - \left(\frac{x}{b}\right)^{\zeta}\right)^{\varsigma}$ | $\varsigma B\left(\varsigma, 1 + \frac{p}{\zeta}\right)b^p$ | |

**Table 3.** Non-central moments and cumulants of common distributions with zero upper-tail index (all moments and cumulants exist).

| Name, Domain | Probability Density or Distribution Function | Moments, $\mu'_p$ | Cumulants, $\kappa_p$ |
|---|---|---|---|
| Poisson $\underline{x} = j, j \in \mathbb{N}_0$ | $f(x) = e^{-\varsigma} \sum\limits_{j=0}^{\infty} \frac{\varsigma^j}{j!} \delta(x - j)$ | | $\varsigma$ |
| Exponential, $x \geq 0$ | $f(x) = e^{-x/\mu}/\mu$ | $p!\mu^p$ | $(p-1)!\mu^p$ |
| Gamma, $\underline{x} \geq 0$ | $f(x) = \frac{(x/\lambda)^{\zeta-1} e^{-x/\lambda}}{\lambda \, \Gamma(\zeta)}$ | $\frac{\Gamma(p+\zeta)}{\Gamma(\zeta)}\lambda^p$ | $\zeta(p-1)!\lambda^p$ |
| Generalized gamma, $\underline{x} \geq 0$ | $f(x) = \frac{1}{\lambda \, \Gamma(\zeta/\varsigma)} \left(\frac{x}{\lambda}\right)^{\zeta-1} \exp\left(-\left(\frac{x}{\lambda}\right)^{\varsigma}\right)$ | $\frac{\Gamma(p/\varsigma+\zeta/\varsigma)}{\Gamma(\zeta/\varsigma)}\lambda^p$ | |
| Weibull, $\underline{x} \geq 0$ | $F(x) = 1 - \exp\left(-\left(\frac{x}{\lambda}\right)^{\zeta}\right)$ | $\Gamma\left(\frac{p}{\zeta} + 1\right)\lambda^p$ | |
| Normal, $\underline{x} \in \mathbb{R}$ | $f(x) = \frac{\exp\left(-\frac{(x-\mu)^2}{2\sigma^2}\right)}{\sqrt{2\pi}\sigma}$ | | $\begin{cases} \mu'_1 = \mu, & p = 1 \\ \sigma^2 & p = 2 \\ 0 & p > 2 \end{cases}$ |
| Half-normal, $\underline{x} \geq 0$ | $f(x) = \frac{2}{\lambda\sqrt{2\pi}} \exp\left(-\frac{x^2}{2\lambda^2}\right)$ | $\frac{2^{p/2}}{\sqrt{\pi}}\Gamma\left(\frac{p+1}{2}\right)\lambda^p$ | |
| Extended half-normal (Chi), $\underline{x} \geq 0$ | $f(x) = \frac{\sqrt{2}}{\lambda \, \Gamma(\zeta/2)} \left(\frac{x^2}{2\lambda^2}\right)^{\frac{\zeta}{2}-\frac{1}{2}} \exp\left(-\frac{x^2}{2\lambda^2}\right)$ | $2^{p/2}\frac{\Gamma\left(\frac{p+\zeta}{2}\right)}{\Gamma\left(\frac{\zeta}{2}\right)}\lambda^p$ | |
| Lognormal ($\ln \underline{x} \sim$ N$(\ln \lambda, \varsigma)$), $\underline{x} \geq 0$ | $f(x) = \frac{\exp\left(-\frac{1}{2\varsigma^2}\left(\ln\left(\frac{x}{\lambda}\right)\right)^2\right)}{\sqrt{2\pi}\,\varsigma x}$ | $e^{\frac{p^2\varsigma^2}{2}}\lambda^p$ | |
| Extreme value type I (EV1), $\underline{x} \in \mathbb{R}$ | $F(x) = \exp\left(-e^{-\frac{x}{\lambda}}\right)$ | | $\frac{(-1)^p \psi^{(p-1)}(1)}{p!}\lambda^p$ |

### 2.3. Second Order Properties

For a stochastic process $\underline{x}(t)$ in continuous time $t$ or $\underline{x}_\tau$ in discrete time $\tau$, we define the cumulative process $\underline{X}(k) \equiv \underline{X}_\kappa$, for continuous time scale $k := \kappa D$, where $\kappa$ denotes discrete time scale, as:

$$\underline{X}(k) \equiv \underline{X}_\kappa := \underline{x}_1 + \underline{x}_2 + \cdots + \underline{x}_\kappa = \int\limits_0^{\kappa D} \underline{x}(t)\, \mathrm{d}t \tag{17}$$

The time average of the original process $\underline{x}_\tau$ for discrete time scale $\kappa$ is

$$\underline{x}_\tau^{(\kappa)} := \frac{\underline{x}_{(\tau-1)\kappa+1} + \underline{x}_{(\tau-1)\kappa+2} + \cdots + \underline{x}_{\tau\kappa}}{\kappa} = \frac{\underline{X}_{\tau\kappa} - \underline{X}_{(\tau-1)\kappa}}{\kappa} \tag{18}$$

The variability of the time-averaged process is quantified by the variance:

$$\gamma_\kappa := \mathrm{var}\left[\underline{x}_\tau^{(\kappa)}\right] \tag{19}$$

This can be extended to a continuous-time process, for which

$$\gamma(k) := \mathrm{var}\left[\frac{\underline{X}(k)}{k}\right], \;\; \gamma_\kappa = \gamma(\kappa D) \tag{20}$$

**Table 4.** Non-central moments of common distributions with upper-tail index $\xi$ (moments and cumulants exist for $p < 1/\xi$). Here, the cumulants do not have simple explicit expressions but can be readily calculated from Equation (9).

| Name, Domain | Probability Density or Distribution Function | Moments, $\mu'_p$ |
|---|---|---|
| Pareto $\underline{x} \geq 0$ | $F(x) = 1 - \left(1 + \xi \frac{x}{\lambda}\right)^{-\frac{1}{\xi}}$ | $B\left(\frac{1}{\xi} - p, p + 1\right)\frac{\lambda^p}{\xi^{p+1}}$ |
| Pareto-Burr-Feller (PBF) $\underline{x} \geq 0$ | $F(x) = 1 - \left(1 + \xi\zeta\left(\frac{x}{\lambda}\right)^{\zeta}\right)^{-\frac{1}{\xi\zeta}}$ | $B\left(\frac{1}{\xi\zeta} - \frac{p}{\zeta}, \frac{p}{\zeta} + 1\right)\frac{\lambda^p}{(\xi\zeta)^{\frac{p}{\zeta}+1}}$ |
| Dagum $\underline{x} \geq 0$ | $F(x) = \left(1 + \frac{1}{\xi\zeta}\left(\frac{x}{\lambda}\right)^{-\frac{1}{\xi}}\right)^{-\xi\zeta}$ | $(\xi\zeta)^{1-\xi p}B(1 - \xi p, \xi p + \xi\zeta)\lambda^p$ |
| Extreme value type II (EV2) $\underline{x} \geq 0$ | $F(x) = \exp\left(-\xi\left(\frac{x}{\lambda}\right)^{-\frac{1}{\xi}}\right)$ | $\Gamma(1 - p\xi)\left(\frac{\lambda}{\xi}\right)^p$ |
| Half Student $\underline{x} \geq 0$ | $f(x) = \frac{2\left(1+\left(\frac{x}{\lambda}\right)^2\right)^{-\frac{1}{2}-\frac{1}{2\xi}}}{\lambda\, B\left(\frac{1}{2}, \frac{1}{2\xi}\right)}$ | $\frac{B\left(\frac{1}{2}+\frac{p}{2}, \frac{1}{2\xi}-\frac{p}{2}\right)}{B\left(\frac{1}{2}, \frac{1}{2\xi}\right)}\lambda^p$ |
| Half extended Student $\underline{x} \geq 0$ | $f(x) = \frac{2\left(\left(\frac{x}{\lambda}\right)^2\right)^{\frac{\zeta}{2}-\frac{1}{2}}\left(1+\left(\frac{x}{\lambda}\right)^2\right)^{-\frac{\zeta}{2}-\frac{1}{2\xi}}}{\lambda\, B\left(\frac{\zeta}{2}, \frac{1}{2\xi}\right)}$ | $\frac{B\left(\frac{1}{2\zeta}+\frac{p}{2}, \frac{1}{2\xi}-\frac{p}{2}\right)}{B\left(\frac{1}{2\zeta}, \frac{1}{2\xi}\right)}\lambda^p$ |
| Generalized beta prime (GBP) $\underline{x} \geq 0$ | $f(x) = \frac{\varsigma\left(\frac{x}{\lambda}\right)^{\varsigma-1}\left(1+\left(\frac{x}{\lambda}\right)^{\varsigma}\right)^{-\frac{\zeta}{\varsigma}-\frac{1}{\xi\varsigma}}}{\lambda\, B\left(\frac{\zeta}{\varsigma}, \frac{1}{\xi\varsigma}\right)}$ | $\frac{B\left(\frac{\zeta}{\varsigma}+\frac{p}{\varsigma}, \frac{1}{\xi\varsigma}-\frac{p}{\varsigma}\right)}{B\left(\frac{\zeta}{\varsigma}, \frac{1}{\xi\varsigma}\right)}\lambda^p$ |

Clearly, this is a function of the time-scale $\kappa$ and is termed the *climacogram* of the process, from the Greek climax (κλίμαξ, meaning scale) [41].

For sufficiently large $k$ (theoretically as $k \to \infty$), we may approximate the climacogram as:

$$\gamma(k) \propto k^{2H-2} \tag{21}$$

where $H$ is termed the Hurst parameter. The theoretical validity of such (power-type) behavior of a process was implied by Kolmogorov (1940) [42,43]. The quantity $2H - 2$ is visualized as the slope of the double logarithmic plot of the climacogram for large time-scales. In a purely random process, $H = 1/2$, while in most natural processes $1/2 \leq H \leq 1$, as first observed by Hurst in 1951 [44]. This natural behavior is known as LRD, (long-term) persistence or Hurst–Kolmogorov (HK) dynamics. A high value of $H$ (approaching 1) indicates enhanced presence of patterns, enhanced change and enhanced uncertainty (e.g., in future predictions). A low value of $H$ (<1/2) indicates enhanced fluctuation or antipersistence.

A stochastic process $\underline{x}(t)$ for which the property (21) is valid not only asymptotically, but precisely for any scale $k$, i.e.,

$$\gamma(k) = \lambda\left(\frac{\alpha}{k}\right)^{2-2H} \tag{22}$$

where $\alpha$ and $\lambda$ are scale parameters with units of time and $\left[x^2\right]$, respectively, is termed the *Hurst–Kolmogorov* (HK) *process* [12].

The HK process is a simple mathematical model offering acceptable approximations for large scales, but it is not physically plausible for small scales because it yields infinite variance of the instantaneous process (as $k \to 0$) [45]. Therefore, *filtered* versions thereof (FHK) with finite variance at all scales are better options to model natural processes. Here we use two versions of FHK, namely:

- The generalized Cauchy-type FHK (FHK-C) with climacogram:

$$\gamma(k) = \lambda_0 \left(1 + (k/\alpha)^{2M}\right)^{\frac{H-1}{M}} \tag{23}$$

- The mixed Cauchy–Dagum-type FHK (FHK-CD) climacogram:

$$\gamma(k) = \lambda_1 \left(1 + \frac{k}{\alpha}\right)^{2H-2} + \lambda_2 \left(1 - \left(1 + \frac{\alpha}{k}\right)^{-2M}\right) \tag{24}$$

In addition to the Hurst parameter $H$, which characterizes the global scaling behavior, when $k \to \infty$, the filtered models include a second scaling exponent $M$ characterizing the local scaling (or smoothness or fractal behavior) when $k \to 0$. Furthermore, the FHK-CD model contains two scale parameters of state, $\lambda_1$ and $\lambda_2$, instead of the single $\lambda$ of the FHK-C, offering greater flexibility.

Once the model climacogram is given, all other second-order properties of the process are uniquely determined through simple mathematical expressions. Thus, the autocovariance function in continuous and discrete time, for lags $h$ and $\eta = h/D$, respectively, is derived from the climacogram through the relationships [2,12]:

$$c(h) := \text{cov}[x(t), \ x(t+h)] = \frac{1}{2} \frac{\text{d}^2 h^2 \gamma(|h|)}{\text{d}h^2} \tag{25}$$

for continuous time and

$$c_\eta := \text{cov}\left[x_\tau, \ x_{\tau+\eta}\right] = \frac{(\eta+1)^2 \gamma_{|\eta+1|} + (\eta-1)^2 \gamma_{|\eta-1|}}{2} - \eta^2 \gamma_{|\eta|} \tag{26}$$

for discrete time, where cov [] stands for covariance.

Finally, the power spectrum $s(w)$ of the process is the Fourier transform of the autocovariance, so that:

$$s(w) := 4 \int_0^\infty c(h) \cos(2\pi wh) \text{d}h \Leftrightarrow c(h) = \int_0^\infty s(w) \cos(2\pi wh) \text{d}w \tag{27}$$

for continuous time and

$$s_\text{d}(\omega) = 2c_0 + 4 \sum_{\eta=1}^\infty c_\eta \cos(2\pi\eta\omega) \Leftrightarrow c_\eta = \int_0^{1/2} s_\text{d}(\omega) \cos(2\pi\omega\eta) \text{d}\omega \tag{28}$$

for discrete time.

### 2.4. Stochastic Simulation

To simulate the discrete-time stochastic process $\underline{x}_\tau$ with any autocovariance function $c_\eta$ we can use the generalized moving average scheme [1,11,12]:

$$\underline{x}_\tau = \sum_{j=-J}^{J} a_j \underline{v}_{\tau-j} \tag{29}$$

where $a_j$ are weights to be calculated from the autocovariance function, $\underline{v}_j$ is white noise averaged in discrete-time (in the general case assumed non-Gaussian) and $J$ is theoretically infinite, so that in all theoretical calculations we assume $J = \infty$, while in the generation case $J$ is a large integer chosen so that the resulting truncation error be negligible.

As explained in [1], the above scheme is opposite to the common schemes of the time series school. Specifically, (a) we use a purely moving average scheme without any

autoregressive term and (b) we do not connect the generating scheme with observations, as the observations have already been used in the model-fitting phase, which is totally isolated from generation. Specifically, the fitting consists of a choice of an appropriate climacogram expression such as (23) or (24) and the estimation of its parameters, as well as the choice of a distribution function, such as those contained in Tables 2–4, and the estimation of its parameters. This tactic assures modelling parsimony. More details on the fitting procedure, which is not covered here, can be found in [2]. Here we only stress the methodological suggestion that we never estimate from data classical moments and cumulants of order greater than 2, because these are *unknowable* from data [31]. While the methodology that we follow heavily depends on high-order moments and cumulants, it is stressed that these are determined by theoretical calculations and never from the data.

Assuming unit variance of the white noise $\underline{v}_j$, writing Equation (29) for $\underline{x}_{\tau+\eta}$, multiplying it by (29) and taking expected values we find the convolution expression for $J = \infty$:

$$c_\eta = \sum_{l=-\infty}^{\infty} a_l a_{\eta+l} \tag{30}$$

We need to find the sequence of $a_\eta$, $\eta = \ldots, -1, 0, 1, \ldots$, so that (30) holds true. The following generic solution of the generating scheme, giving the coefficients $a_\eta$, has been proposed by Koutsoyiannis [1]:

$$a_\eta = \int_{-1/2}^{1/2} e^{2\pi i(\vartheta(\omega)-\eta\omega)} A^{\mathrm{R}}(\omega)\mathrm{d}\omega \tag{31}$$

where $\mathrm{i} := \sqrt{-1}$, $\vartheta(\omega)$ is any (arbitrary) odd real function (meaning $\vartheta(-\omega) = -\vartheta(\omega)$) and

$$A^{\mathrm{R}}(\omega) := \sqrt{2s_{\mathrm{d}}(\omega)} \tag{32}$$

As proved by Koutsoyiannis [1], the sequence of $a_\eta$:

1. Consists of real numbers, despite the expression in (31) involving complex numbers;
2. Satisfies precisely Equation (30); and
3. Is easy and fast to calculate using the fast Fourier transform (FFT).

This theoretical result is readily converted into a numerical algorithm, which consists of the following steps [1]:

1. From the continuous-time stochastic model, expressed through its climacogram $\gamma(k)$, we calculate its autocovariance function in discrete time (assuming time step $D$) by Equation (26). (This step is obviously omitted if the model is already expressed in discrete time through its autocovariance function).
2. We choose an appropriate number of coefficients $J$ that is a power of 2 and perform inverse FFT (using common software) to calculate the discrete-time power spectrum and the frequency function $A^{\mathrm{R}}(\omega)$ for an array of $\omega_j = jw_1$, $j = 0, 1, \ldots, J$, $w_1 := 1/JD$:

$$s_{\mathrm{d}}(\omega_j) = 2c_0 + 4\sum_{\eta=1}^{J} c_\eta \cos(2\pi\eta\omega_j), \quad A^{\mathrm{R}}(\omega_j) = \sqrt{2s_{\mathrm{d}}(\omega_j)} \tag{33}$$

3. We choose $\vartheta(\omega)$ (see below) and we form the arrays (vectors) $A^{\mathrm{R}}$ and $A^{\mathrm{I}}$, both of size $2J$ indexed as $0, \ldots, 2J - 1$, with the superscripts R and I standing for the real and imaginary part of a vector of complex numbers, respectively:

$$\left[A^{\mathrm{R}}\right]_j = \begin{cases} A^{\mathrm{R}}(\omega_j)\cos(2\pi\vartheta(\omega_j))/2, & j = 0, \ldots, J \\ \left[A^{\mathrm{R}}\right]_{2J-j}, & j = J+1, \ldots, 2J-1 \end{cases} \tag{34}$$

$$\left[ A^{\mathrm{I}} \right]_j = \begin{cases} -A^{\mathrm{R}}(\omega_j) \sin\left(2\pi\vartheta(\omega_j)\right)/2, & j = 0,\ldots,J-1 \\ 0 & j = J \\ -\left[ A^{\mathrm{I}} \right]_{2J-j} & j = J+1,\ldots,2J-1 \end{cases} \tag{35}$$

4. We perform FFT on the vector $A^{\mathrm{R}} + \mathrm{i}\, A^{\mathrm{I}}$ (using common software), and get the real part of the result, which is precisely the sequence of $a_\eta$.

By choosing $J$ as a power of 2, the vectors $A^{\mathrm{R}}$ and $A^{\mathrm{I}}$ will have size $2J$ which is also a power of 2, thus maximizing the speed of the FFT calculations. (More details are contained in a supplementary file in [1], which includes numerical examples along with the simple code needed to do these calculations on a spreadsheet).

Remarkably, Equation (31) gives, instead of a single solution, a family of infinitely many solutions. All of them preserve exactly the second-order characteristics of the process and each of them is characterized by the chosen function $\vartheta(\omega)$. Even assuming $\vartheta(\omega) = \vartheta_0 \mathrm{sign}\omega$ with constant $\vartheta_0$, again there are infinitely many solutions, each one characterized by the value of $\vartheta_0$. Also, even if the sequence of $\vartheta(\omega_j)$ is constructed as a sequence of random numbers, again Equation (30) will be satisfied and the resulting $a_\eta$ can be directly used in generation. The availability of infinitely many solutions enables preservation of additional statistics, such as those related to time asymmetry [1,27].

The special case $\vartheta(\omega) = 0$ gives a symmetric solution with respect to positive and negative $\eta$:

$$A^{\mathrm{S}}(\omega) \equiv A^{\mathrm{R}}(\omega) = \sqrt{2s_{\mathrm{d}}(\omega)}, \quad a_j^{\mathrm{S}} = \int\limits_0^{1/2} \sqrt{2s_{\mathrm{d}}(\omega)} \cos(2\pi j\omega)\mathrm{d}\omega = a_{-j}^{\mathrm{S}} \tag{36}$$

where the superscript S stands for symmetric. This has been known as the *symmetric moving average* (SMA) scheme [11], while any other solution denotes an *asymmetric moving average* (AMA) scheme.

In addition, there exist several options related to the distribution of the white noise $\underline{v}_\tau$, which in general is not Gaussian. Hence, preservation of moments and cumulants of any order becomes possible. Specifically, by virtue of Equation (13), the $p$th cumulants $\kappa_p$ and $\kappa_p^{(v)}$ of the processes $\underline{x}_\tau$ and $\underline{v}_\tau$, respectively, are related by:

$$\kappa_p = \sum_{j=-J}^{J} a_j^p \, \kappa_p^{(v)} \tag{37}$$

Solving for $\kappa_p^{(v)}$ we find:

$$\kappa_p^{(v)} = \frac{\kappa_p}{\sum_{l=-J}^{J} a_j^p} \tag{38}$$

Given the so-calculated $\kappa_p^{(v)}$ for any order $p$, the distribution function of the white noise is fully determined.

### 2.5. Distribution Function Approximation

A problem usually met in practice, including in the present simulation framework, is to approximate a distribution function up to an order $p_{\max}$. A convenient way to make the approximation is to choose a number $L$ of elementary distribution functions from Tables 2–4, thus, defining the white-noise processes $\underline{w}_l$, $l = 1,\ldots,L$, and obtaining the approximation $\underline{v}_\tau'$ of $\underline{v}_\tau$ as a linear combination of $\underline{w}_l$ with weights $a_l'$, i.e.,:

$$\underline{v}_\tau' = \sum_{l=1}^{L} a_l' \underline{w}_l \tag{39}$$

The cumulants $\kappa_p^{(w_l)}$ of $\underline{w}_l$ are then determined from Tables 2–4 and those of $\underline{v}'_\tau$, by virtue of (13), are:

$$\kappa_p^{(v')} = \sum_{l=1}^{L} a_l'^p \, \kappa_p^{(w_l)} \tag{40}$$

The goodness of the approximation up to order $p_{\max}$ is given by an error expression such as:

$$e_1 := \sum_{p=2}^{p_{\max}} \left( \left(\kappa_p^{(v')}\right)^{\frac{1}{p}} - \left(\kappa_p^{(v)}\right)^{\frac{1}{p}} \right)^2, \quad e_2 := \sum_{p=2}^{p_{\max}} \left( \frac{1}{p} \ln\left( \frac{\kappa_p^{(v')}}{(\kappa_p^{(v)})} \right) \right)^2 \tag{41}$$

where the second form ($e_2$) is more appropriate if all cumulants are positive and increasing fast. In order for the above equations to work in all cases, even when $\kappa_p$ is negative and $p$ is even, the quantity $(\kappa_p)^{1/p}$ is meant to denote the quantity $\text{sign}(\kappa_p) \, |\kappa_p|^{1/p}$; this convention is followed throughout the entire paper. By minimizing either $e_1$ or $e_2$ using a common solver, we simultaneously find the series of weights $a_l'$ and the parameters of the marginal distribution of each of $\underline{w}_l$. Further details will be given in the applications of Section 3, where it will also be seen that, for a sufficient approximation, the number of constituent distributions $L$ of $\underline{w}_l$ is small, usually 1 or 2.

It is stressed that, in each of the above error expressions, we have intentionally excluded the error of the cumulants of order 1, i.e., the mean values. Therefore, we expect that with this procedure the mean will not be preserved. However, this can be easily tackled by adding a constant $c$ to $\underline{v}'_\tau$. Apparently, the required shift should be

$$c = \kappa_1^{(v)} - \kappa_1^{(v')} \tag{42}$$

Based on the above approximation, the generation process will produce the stochastic process

$$\underline{x}'_\tau := \sum_{j=-J}^{J} a_j \underline{v}'_{\tau-j} \tag{43}$$

where, if the approximation is satisfactory, we reasonably expect that the statistical properties of $\underline{x}'_\tau$ will be equal to those of $\underline{x}_\tau$. This proves to be always the case if the domain of the stochastic variable $\underline{x}_\tau$ is unbounded in both directions (i.e., $\underline{x}_\tau \in \mathbb{R}$), but some additional manipulation (post-processing) may be needed if the domain of $\underline{x}_\tau$ is not the entire real line, or if the distribution function of $\underline{x}_\tau$ has discontinuities, as will be illustrated in the applications of the next section.

## 3. Applications and Results

We illustrate the methodology by five applications for bounded $\underline{x}_\tau$ as this case is more demanding (the unbounded case is much easier). Three applications are synthetic mathematical examples used as benchmarks, namely the exponential distribution, which is bounded from below, and the uniform distribution, which is bounded from both below and above. The next two are real-world applications dealing with one of the most challenging natural processes, namely the precipitation process, which is bounded from below (by 0), highly intermittent, and with heavy distribution tail. The latter two applications refer to two different time scales, fine (hourly) and coarse (annual). In the synthetic example with the exponential distribution and in the two real-world applications, the stochastic processes are persistent with a large Hurst parameter, ranging from 0.80 to 0.92. In the synthetic examples of the uniform distribution, we use both a persistent and an antipersistent process, with Hurst parameters 0.70 and 0.20, respectively.

### 3.1. Simulating a Persistent Process with Exponential Distribution

For a process with exponential distribution, which is a subcase of the gamma distribution, there exist generation algorithms for the case of short-range (Markov) dependence

(e.g., [46]). As already mentioned, a more general algorithm for generation of any type of marginal distribution has recently been proposed by Lombardo et al. [28], but again under the condition of the Markov dependence. However, the method proposed here can generate such a process irrespective of the type of the dependence, whether SRD or LRD.

For illustration we assume an FHK-C model (Equation (23)) with parameters $H = 0.8$, $M = 0.5$, $\alpha = 1, \lambda_0 = 1.32$, so that $\gamma_1 = 1$. The FHK-C climacogram is shown in Figure 1b, marked as "theoretical", while the resulting autocorrelation function is shown in Figure 1c. As in the exponential distribution (from Table 3), $\mu = \sqrt{\gamma_1} = 1$, the cumulants of the process $\underline{x}_\tau$ are $\kappa_p = (p-1)!$. These are depicted in Figure 1a, along with the cumulants of $\underline{v}_\tau$ determined from Equation (38), where, to avoid big numbers, the quantities $\kappa_p^{1/q}$ are plotted. The coefficients $a_j$, needed to evaluate $\kappa_p^{(v)}$ in Equation (38), are determined from the SMA (symmetric) generation scheme (Equation (36)) with $J = 1024$.

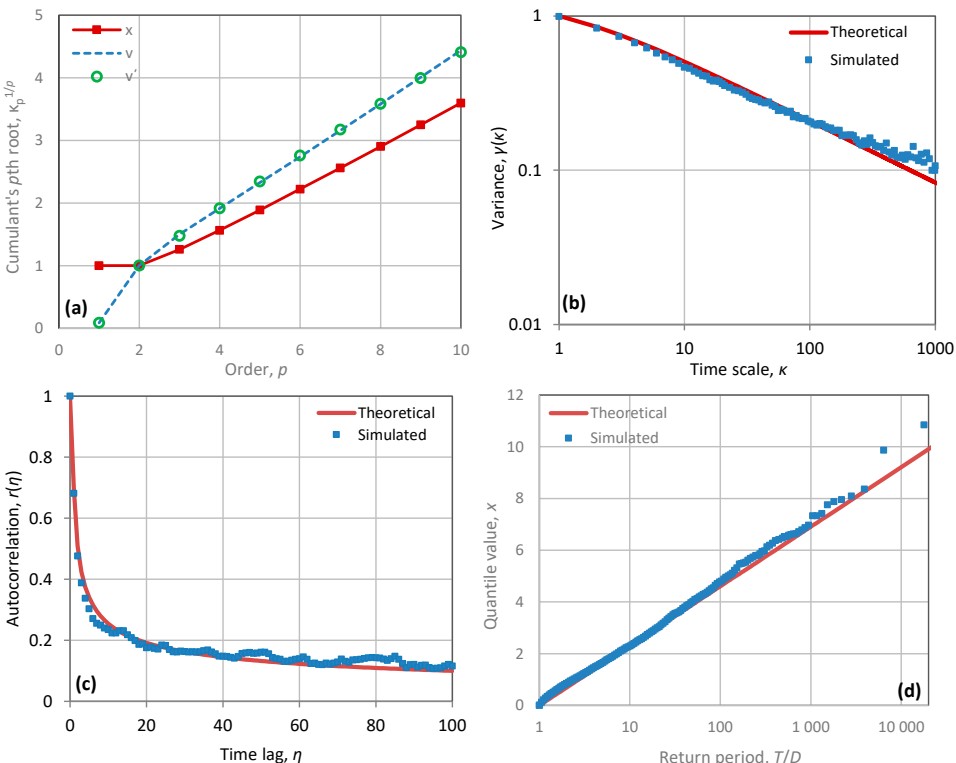

**Figure 1.** Graphical depiction of the results of the simulation application for a synthetic example of a persistent FHK process with exponential distribution: (**a**) cumulants; (**b**) climacogram; (**c**) autocorrelogram; (**d**) marginal distribution.

Coming to the approximation $\underline{v}'_\tau$ of $\underline{v}_\tau$, we use two constituents $\underline{w}_l$ with gamma distributions and allow a discontinuity $P_l$ at $w = 0$ in each of them. Assuming unit variance in each of them, from the equations of Table 3 we have $\zeta \lambda^2 = 1$, so that the continuous part of the distribution is fully determined by the shape parameter $\zeta$. Hence, the approximation $\underline{v}'_\tau$, according to Equation (39), is determined by the parameters $\zeta_1, \zeta_2, P_1, P_2, a'_1, a'_2$, which are calculated by minimizing $e_2$ in Equation (41), assuming $p_{\max} = 10$. The resulting values of the parameters are $\zeta_1 = 1.255, \zeta_2 = 30, P_1 = 0.298, P_2 = 1, a'_1 = 1.333, a'_2 = -0.0655$, while the required shift of Equation (42) is negligible ($c \approx 0$). The cumulants of $\underline{v}'_\tau$ are also plotted in Figure 1a, where it can be seen that they are indistinguishable from those of $\underline{v}_\tau$ and thus the achieved approximation is very good.

The generation of values of $\underline{v}'_\tau$ is quite easy using a random number generator for the gamma distribution. From a series of random numbers $v'_\tau$, a total of $n = 10,000$ values of $\underline{x}_\tau$ are then determined from Equation (29). A small number (6.6%) of them are small negative values. To remedy this problem, we reflect these values about zero, or, in other

words, replace $x_\tau$ with $-x_\tau$. Theoretically, this remedy will have a distorting effect in the multivariate distribution of $\underline{x}_\tau$, but in fact, this effect turns out to be negligible.

Comparison of the theoretical statistical characteristics of the distribution of $\underline{x}_\tau$ to the empirical ones of the generated sample are shown in the panels of Figure 1. In the empirical climacogram (Figure 1b), the plotted points correspond to unbiased estimates of variance; this is achieved by adding the quantity $\gamma(n) = 0.0331$ to the classical statistical estimates, as explained in [2]. The empirical climacogram agrees well with the theoretical one. The empirical autocorrelation is shown in Figure 1c. Here, the bias correction was applied using an approximate method from [47], according to which the unbiased estimate is the weighted sum of the classical autocorrelation estimate and the number 1, with the weight of the latter being equal to $1/n'$, where $n' := \gamma(1)/\gamma(n)$ is the so-called *equivalent sample size* of any process, and differs substantially from $n$ if the process is persistent [48]. (We note that a precisely unbiased estimate of autocovariance has been provided by [49] but this is more laborious). Finally, Figure 1d shows a comparison of the theoretical and empirical marginal distribution of $\underline{x}_\tau$. The empirical distribution of each value of the generated time series, arranged in ascending order, so that $x_{(i:n)}$ be the $i$th smallest value of the series of $n$ values, was estimated on the basis of unbiasedness of the logarithm of return period $T_{(i:n)}$. As shown in [2], this estimate is

$$\frac{T_{(i:n)}}{D} = \frac{n + e^{1-\gamma} - 1}{n - i + e^{-\gamma}} = \frac{n + 0.526}{n - i + 0.561} \tag{44}$$

Again, the agreement between theoretical and the empirical distributions is very good.

For comparison, a conventional method using an ARMA(1,1) model and a normalizing transformation is given in the Appendix A for the same case study.

### 3.2. Simulating a Persistent Process with Uniform Distribution

The simulation of a persistent process with uniform distribution is more demanding because of the double boundedness and the sharp discontinuities of the density function at the bounds, while linear generation procedures tend to generate unbounded processes with smooth density. On the other hand, the double boundedness offers an option of approximation with a process $\underline{v}'_\tau$ that takes on a finite number of values. In other words, we assume that the stochastic variable $\underline{v}'_\tau$ is discrete, taking on values $v'_i$ with probabilities $P_i$, as illustrated in Figure 2. The details of this approximation will be explained in a while. Despite $v'_\tau$ being assumed discrete, thanks to the fact that the generation of $\underline{x}_\tau$ via Equation (29) involves a linear combination of very many variables $v'_\tau$, the variable $\underline{x}_\tau$ will in effect be continuous.

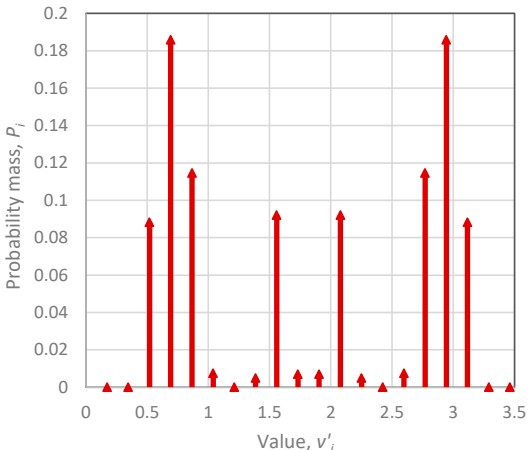

**Figure 2.** Probability mass function of the discretized white noise used in the simulation application for a synthetic example of a persistent FHK process with uniform distribution.

As in the previous case, for illustration, we assume an FHK-C model (Equation (23)) with $\gamma_1 = 1$. We note that the fourth cumulant of this uniform distribution, which in this case equals the coefficient of kurtosis, is $\kappa_4 = -1.2$. The fourth cumulant of $\underline{v}_\tau$ ($\kappa_4^{(v)}$) should necessarily be lower than that ($\kappa_4^{(v)} < -1.2$) for a persistent process. On the other hand, it is known than the kurtosis of any distribution cannot be lower than $-2$. Therefore, the margin for having a positively autocorrelated process $\underline{x}_\tau$ with uniform distribution is rather small. An FHK-C model with parameters $H = M = 0.7$, $\alpha = 1, \lambda_0 = 1.346$ (so that $\gamma_1 = 1$) yields a feasible $\kappa_4^{(v)} = -1.76$, while, for instance, the case $H = M = 0.75$, would yield an infeasible $\kappa_4^{(v)} = -2.02$. The FHK-C climacogram for the feasible parameter set ($H = M = 0.7$) is shown in Figure 3b, marked as "theoretical", while the resulting autocorrelation function is shown in Figure 3c. In order for the uniform distribution to have variance $\gamma_1 = 1$, its upper bound should be $b = \sqrt{12} = 3.464$, with lower bound $a = 0$. The cumulants of the process $\underline{x}_\tau$, determined from Table 2 and Equation (9), are shown in Figure 3a, along with the cumulants of $\underline{v}_\tau$ determined from Equation (38) (for the convention used for $\kappa_p^{1/q}$ for negative quantities and $p$ even, see the note in Section 2.4 below Equation (41)). The coefficients $a_j$, needed to evaluate $\kappa_p^{(v)}$ in Equation (38), are determined from the SMA (symmetric) generation scheme (Equation (36)) with $J = 1024$.

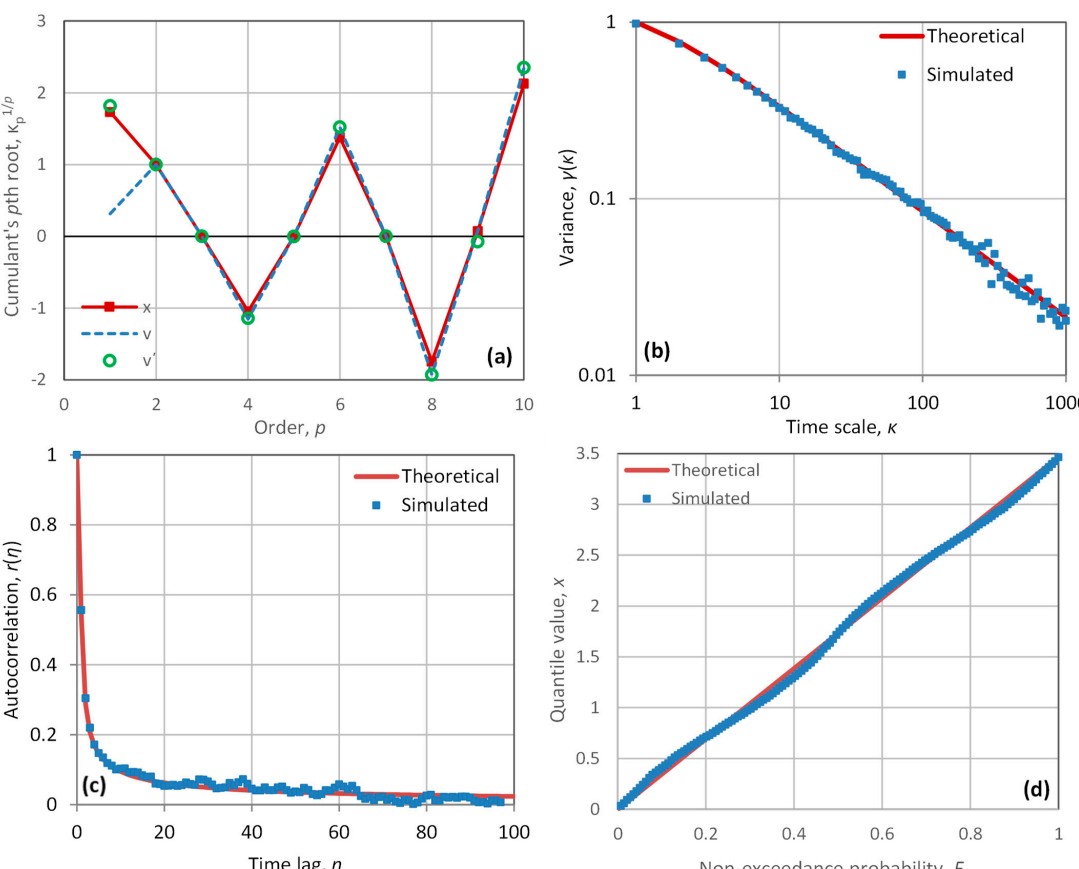

**Figure 3.** Graphical depiction of the results of the simulation application for a synthetic example of a persistent FHK process with uniform distribution: (**a**) cumulants; (**b**) climacogram; (**c**) autocorrelogram; (**d**) marginal distribution.

Comparisons of the theoretical statistical characteristics of the distribution of $\underline{x}_\tau$ to the empirical ones of the generated sample are shown in the panels of Figure 3, which are similar as those in Figure 1. A difference is that in panel (d), instead of estimating the return period of each $x_{(i:n)}$ (the $i$th smallest value of the series of $n$ values), we give the

non-exceedance probability $F(x)$, estimated on the basis of its unbiasedness. In this case, the unbiased estimate is [2]:

$$F\left(x_{(i:n)}\right) = \frac{i}{n+1} \tag{45}$$

The approximation $\underline{v}'_\tau$ of $\underline{v}_\tau$ is done through the discretization of the former described above. Twenty equidistant $v'_i$ with probabilities $P_i$ are assumed, where $v'_i = i/b$, $i = 1, \ldots, 20$. The distribution of $v'_i$ was assumed symmetric, i.e., $P_i = P_{21-i}$, so that the unknown parameters to be optimized are ten, namely, $P_1, \ldots, P_{10}$. These are calculated by minimizing $e_1$ in Equation (41), assuming $p_{\max} = 10$. The resulting values are shown graphically in Figure 2. It is remarkable that the distribution of $\underline{v}'_\tau$ is far from uniform, despite the fact that the cumulants of $\underline{v}'_\tau$, as seen in Figure 3a, are not very different from those of $\underline{x}_\tau$, which has uniform distribution. The cumulants of $\underline{v}'_\tau$, also plotted in Figure 1a, are indistinguishable from those of $\underline{v}_\tau$; thus, the achieved approximation is very good. An exception is seen in the first cumulants of $\underline{v}'_\tau$ and $\underline{v}_\tau$, which are quite different; thus, the required shift of Equation (42) is not negligible, namely $c = -1.503$.

The generation phase is quite easy, as values of $\underline{v}'_\tau$ are readily generated by inverse-transform sampling, given the staircase-like distribution function of a discrete stochastic variable. A total $n = 10\,000$ values of $\underline{x}_\tau$ are then generated from Equation (29). A small number (~2%) of them are either small negative values or somewhat greater than $b$. As in the previous case, we reflect the negative values about zero, replacing $x_\tau$ with $-x_\tau$. Likewise, we reflect the very high values about $b$, replacing $x_\tau$ with $2b - x_\tau$.

In all panels of Figure 3, the agreement between theoretical and the empirical characteristics is very good.

### 3.3. Simulating an Antipersistent Process with Uniform Distribution

For further illustration, we examine the same uniform distribution as above but for an antipersistent process (with $H < 1/2$). Actually, this case is easier as the changes in kurtosis is smaller than in the previous case; thus, feasibility of the solution is assured.

Again, an FHK-C model was assumed, now with parameters $H = 0.2$, $M = 0.8$, $\alpha = 1, \lambda_0 = 2$ (so that $\gamma_1 = 1$, while $\kappa_4^{(v)} = -1.265$). All other choices are the same as in the previous application (e.g., upper bound $b = \sqrt{12} = 3.464$, etc.) The approximation $\underline{v}'_\tau$ of $\underline{v}_\tau$ through discretization is depicted in Figure 4. Again, this differs substantially from the uniform distribution, even though the cumulants of $\underline{v}'_\tau$, as seen in Figure 5a, are virtually indistinguishable from those of $\underline{x}_\tau$ and $\underline{v}_\tau$. Yet there is a substantial difference in the first cumulants of $\underline{v}'_\tau$ and $\underline{v}_\tau$, so that the required shift of Equation (42) is large, $c = 13.675$.

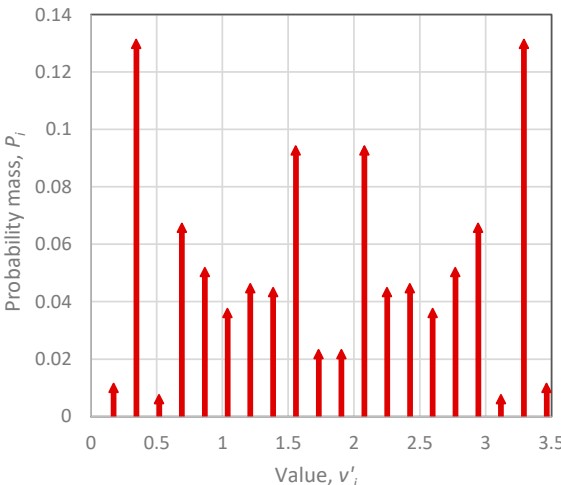

**Figure 4.** Probability mass function of the discretized white noise used in the simulation application for a synthetic example of an antipersistent FHK process with uniform distribution.

Comparisons of the theoretical statistical characteristics of the distribution of $\underline{x}_\tau$ to the empirical ones of the generated sample are shown in the panels of Figure 5. In all panels the agreement between theoretical and the empirical characteristics is very good.

### 3.4. Simulating the Precipitation Process at the Hourly Time Scale

Here we use a recently developed [2] full stochastic model of the precipitation process at any time scale *k*. This model gives directly the ombrian relationships (else known as intensity-duration-frequency curves) but it also provides any stochastic characteristic of the precipitation process that is required for stochastic simulation. Furthermore, in [2] this model has been applied to construct the ombrian curves by fitting the model in some locations, but the model was not used for stochastic simulation. Among the locations studied in [2], here we provide a stochastic simulation for rainfall in Bologna, using the parameter values fitted there. The application in this subsection is for the hourly scale, while an additional application for the annual scale is given in the next subsection.

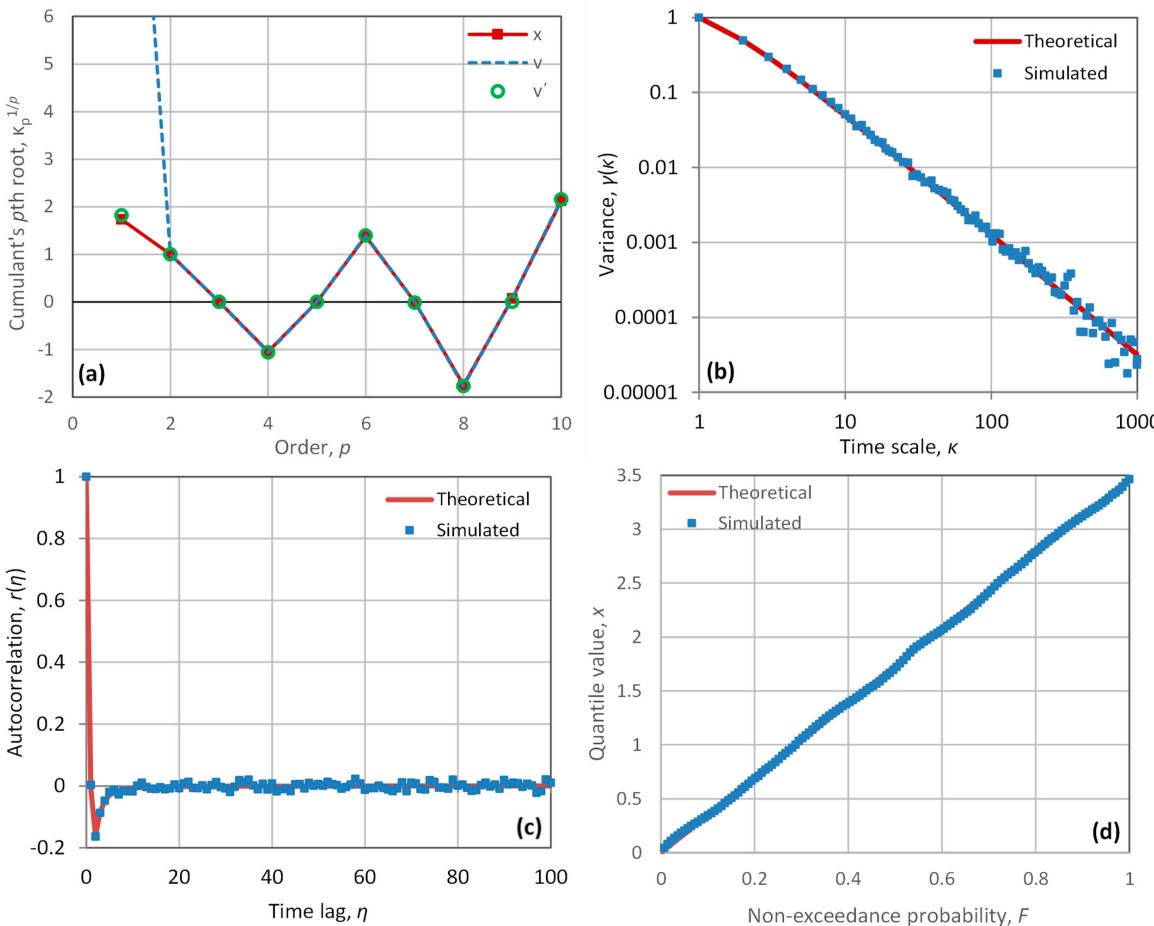

**Figure 5.** Graphical depiction of the results of the simulation application for a synthetic example of an antipersistent FHK process with uniform distribution: (**a**) cumulants; (**b**) climacogram; (**c**) autocorrelogram; (**d**) marginal distribution. Notice in panel (**a**) that the first cumulant of $\underline{v}$ is out of the graph area as it is very large ($\kappa_1^{(v)} = 15.49$).

The model is based on the following assumptions, which are mathematically consistent (with one exception as detailed below):

1. Pareto distribution with discontinuity at the origin for small time scales (Table 5, Equation (46), left). The tail index $\xi$ is constant for all time scales *k*, while the probability wet, $P_1^{(k)}$, and the state scale parameter, $\lambda(k)$, are functions of the time scale *k*.
2. Continuous PBF distribution, possibly with discontinuity at zero, for large time scales (Table 5, Equation (46), right). In this case, a new parameter $\zeta(k)$ is introduced,

which is again a function of time scale. The Pareto distribution is a special case of the PBF for $\zeta(k) = 1$. In contrast to the Pareto distribution, whose density is a consistently decreasing function of $x$, the PBF tends to be bell-shaped for increasing $\zeta(k)$, a property consistent with empirical observation and reason.

3.　Constant mean $\mu$ of the time-averaged process.

4.　Climacogram of type FHK-CD (Equation (24)), where to reduce the number of parameters it is assumed that $M = 1 - H$, thus getting Equation (48) in Table 5. By inspection of Equation (48), it is seen that, as $k \to \infty$, $\gamma(k) \to 0$, which makes the process ergodic; for $k = 0$, $\gamma(0) = \gamma_0 = \lambda_1 + \lambda_2$, which is finite, as required for physical consistency.

5.　Probability wet and dry, $P_1^{(k)} = 1 - P_0^{(k)}$, varying with time scale according to Equation (49) in Table 5. It is clarified that two different expressions are used for the small and the large scales, where the transition time scale from the Pareto to the PBF distribution is denoted as $k^*$. In the Pareto case, $P_1^{(k)}$ can be determined directly from the climacogram and the mean (left column of Equation (49) in Table 5). For the PBF case, an additional equation is required, which has been derived based on maximum entropy considerations [50] and involves an additional parameter $\theta$ ($0 \leq \theta \leq 1$). Continuity of the transition demands that $\zeta(k^*) = 1$.

**Table 5.** Mathematical relationships of the ombrian model. The ombrian curves per se are given in the last row.

| Quantity and Symbol | Small Scales, $k \leq k^*$ **(Pareto)** | Large Scales, $k \geq k^*$ **(PBF)** | Eqn. no. |
|---|---|---|---|
| Distribution function, $F^{(k)}(x)$ | $1 - P_1^{(k)}\left(1 + \xi\frac{x}{\lambda(k)}\right)^{-1/\xi}$ | $1 - P_1^{(k)}\left(1 + \xi'\zeta(k)\left(\frac{x}{\lambda(k)}\right)^{\zeta(k)}\right)^{-\frac{1}{\xi'\zeta(k)}}$ | (46) |
| Mean, $\mathrm{E}\left[\underline{x}^{(k)}\right]$ | | $\mu$ | (47) |
| Climacogram, $\gamma(k)$ | $\lambda_1\left(1 + \frac{k}{\alpha}\right)^{2H-2} + \lambda_2\left(1 - \left(1 + \frac{\alpha}{k}\right)^{2H-2}\right)$ | | (48) |
| Probability wet, $P_1^{(k)}$ | $\frac{1-\xi}{1/2-\xi}\frac{\mu^2}{\gamma(k)+\mu^2}$ | $1 - \left(1 - P_1^{(k^*)}\right)^{(k/k^*)^\theta}, \ (0 \leq \theta \leq 1)$ | (49) |
| Lower tail index (inverse), $\frac{1}{\zeta(k)}$ | $1$ | $\sqrt{(1 - 2\xi)\left(P_1^{(k)}(\gamma(k)/\mu^2 + 1) - 1\right)}$ | (50) |
| Upper tail index, $\xi$ | $\xi$ | $\xi' = \frac{\xi}{\zeta(k)}$ | (51) |
| Scale parameter (inverse), $\frac{1}{\lambda(k)}$ | $\frac{P_1^{(k)}}{\mu(1-\xi)}$ | $\frac{P_1^{(k)}}{\mu}\left(1 + \frac{1}{(1-\xi)(\zeta(k))^2} - \frac{1}{(\zeta(k))^{\sqrt{2}}}\right)$ | (52) |
| Quantile, $x$ | $\lambda(k)\frac{\left(P_1^{(k)}T/k\right)^\xi - 1}{\xi}$ | $\lambda(k)\left(\frac{\left(P_1^{(k)}T/k\right)^\xi - 1}{\xi}\right)^{\frac{1}{\zeta(k)}}$ | (53) |

Both the decreasing (Pareto) and the bell-shaped (PBF) types of probability densities are consistent with natural behaviors for small and large time scales, respectively. It can be seen that the tail index of the PBF distribution in the form in Table 4, is not $\xi$ but $\xi' = \xi/\zeta(k)$ and tends to zero as $k \to \infty$. For large time scales, this violates a requirement of a constant tail index, which is theoretically justified in [2]. The alternative to keep a constant tail index $\xi$ would result in a finite variance as $k \to \infty$ (with a coefficient of variation $\xi/\sqrt{1 - 2\xi}$), i.e., in a nonergodic process, which clearly is not an option in stochastic simulation.

To complete the model, the functions $\lambda(k)$ and $\zeta(k)$ should be determined from the mean $\mu$ and the climacogram $\gamma(k)$. This has been done in [2] and the results are shown in Table 5. The final relationships rely on the mean $\mu$, the climacogram $\gamma(k)$, the probability

wet $P_1^{(k)}$ and the tail index $\xi$. For the precipitation process in Bologna, the following model parameters have been estimated in [2], while the transition time scale was set $k^* = 96$ h:

- Mean intensity, $\mu = 0.0823$ mm/h;
- Intensity scale parameters, $\lambda_1 = 0.00110$ mm²/h², $\lambda_2 = 1.43$ mm²/h²;
- Time scale parameter, $\alpha = 8.74$ h;
- Hurst parameter, $H = 0.92$; fractal (smoothness) parameter, $M = 1 - H = 0.08$;
- Exponent of the expression of probability dry/wet, $\theta = 0.787$;
- Upper tail index, $\xi = 0.121$.

For the hourly time scale, the resulting distribution is Pareto (Tables 4 and 5) with a discontinuity at zero, $P_0 := P\{x = 0\} = 1 - P_1$ and parameters $\xi = 0.121$, $\lambda = 2.046$ mm/h, $P_1 = 0.0354$. The FHK-CD climacogram is shown in Figure 6b (marked as "theoretical"), while the resulting autocorrelation function is shown in Figure 6c. The cumulants of the process $\underline{x}_\tau$ are shown in Figure 6a, along with the cumulants of $\underline{v}_\tau$ determined from Equation (38). The coefficients $a_j$, needed to evaluate $\kappa_p^{(v)}$ in Equation (38), are determined from an AMA (asymmetric) generation scheme (Equation (31)) with $J = 1024$ and phases $\vartheta$ generated randomly (this contributes to a realistic shape of generated rainfall events).

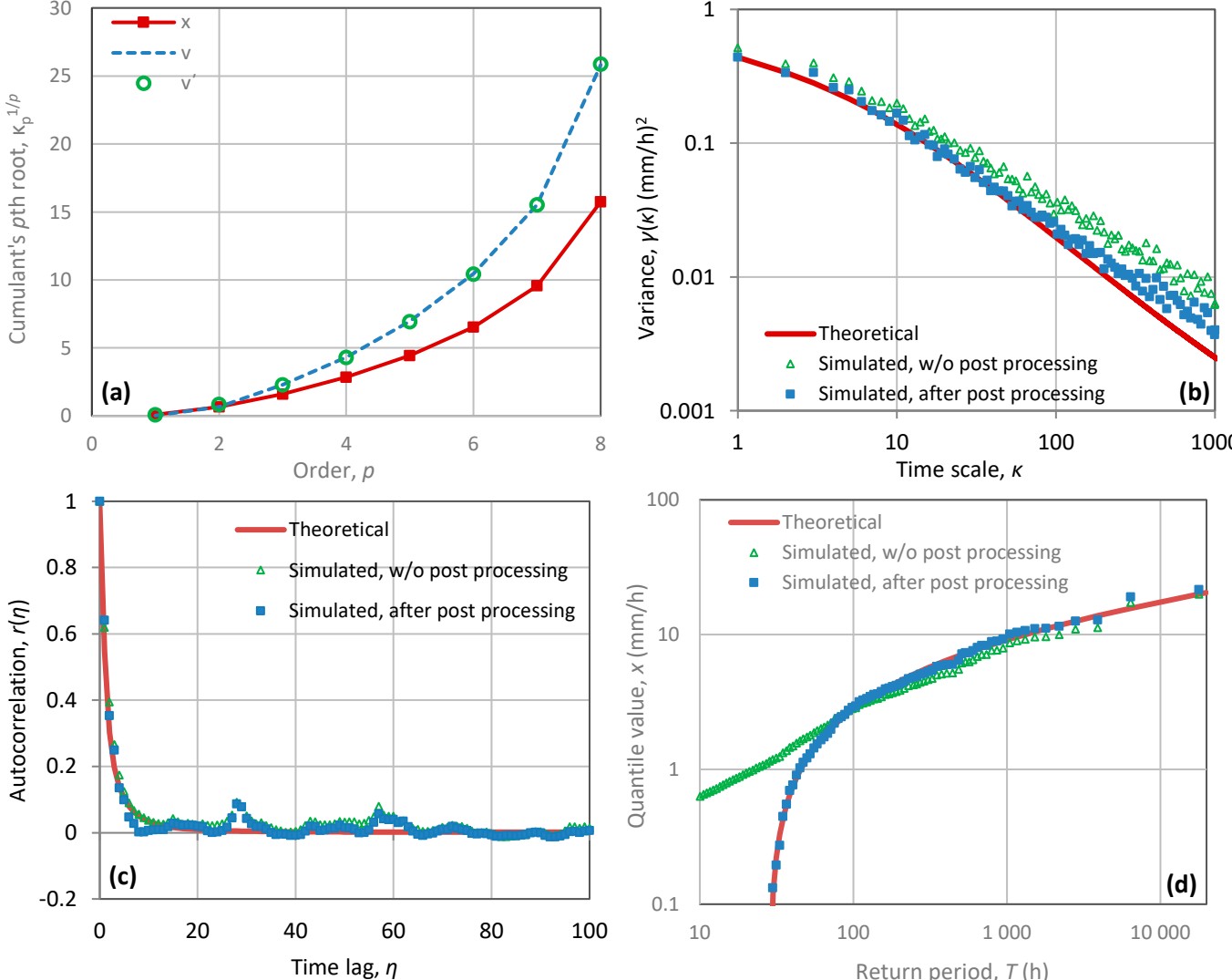

**Figure 6.** Graphical depiction of the results of the simulation application for a real-world case study for the precipitation process in Bologna at the hourly time scale, modelled as a persistent FHK process with Pareto distribution with discontinuity at zero: (**a**) cumulants; (**b**) climacogram; (**c**) autocorrelogram; (**d**) marginal distribution.

For the approximation $\underline{v}'_\tau$ of $\underline{v}_\tau$, we use a single Pareto distribution and allow a discontinuity $P_1$ at $v' = 0$. For mathematical consistency, the tail index of $\underline{v}'_\tau$ should necessarily be $\xi = 0.121$, so that the moments of order beyond $1/\xi = 8.2$ be infinite as is the case with the moments of $\underline{x}_\tau$. The other parameters of the Pareto distribution of $\underline{v}'_\tau$ are calculated by minimizing $e_2$ in Equation (41), setting $p_{\max} = 8$, and are found to be $\lambda^{(v')} = 3.681$, $P_1^{(v')} = 0.0171$, while the required shift of Equation (42) is negligible ($c = 0$). The cumulants of $\underline{v}'_\tau$ are also plotted in Figure 6a, where it can be seen that they are indistinguishable from those of $\underline{v}_\tau$ and thus the achieved approximation is very good.

Because of the very small value of $P_1^{(v')}$, a very large number of $\underline{v}'_\tau$ (98.3%) will be zero. The nonzero values will determine the locations of rainfall events, i.e., sequences of non-zero $x_\tau$. It is not reasonable to make these locations purely random and for this reason we devised the following procedure. A first model run is done with $P_1^{(v')} = 1$ (no zeros). Subsequently, we find a threshold $c_0$ so that the fraction of values $x_\tau$ that are greater than $c_0$ equal $P_1^{(v')}$. In a second model run we set $v'_\tau = 0$ at those $\tau$ where in the first run $x_\tau < c_0$. For the remaining $\tau$, we generate $v'_\tau$ from the continuous part of $\underline{v}'_\tau$. This procedure allows clustering of the precipitation events, as typically happens in reality.

The values $x_\tau$ in the second run will unavoidably be nonzero, because the generating Equation (29) involves a linear combination of very many $v'_\tau$ and this can hardly result in zero values. Therefore, post-processing of the generated time series is required, in order to reinstate the required number of zeros. This consists of replacing $x_\tau$ by $x'_\tau$, determined as:

$$x'_\tau = \begin{cases} 0, & x_\tau < c_1 \\ l(x_\tau - c_1)^m, & x_\tau \geq c_1 \end{cases} \tag{54}$$

where $c_1, l$ and $m$ are the parameters of the post-processing phase. These are determined by minimizing the total error (in effect making it zero) in preserving the probability wet, and the first and second cumulants of the distribution. In our application, the post-processing parameters have been found to be $c_0 = 3.18\,\text{mm/h}$, $c_1 = 1.15\,\text{mm/h}$, $l = 1.877$, $m = 0.832$.

Comparisons of the theoretical statistical characteristics of the distribution of $\underline{x}_\tau$ to the empirical ones of the generated sample, both before and after post-processing, are shown in the panels of Figure 6. The empirical climacogram is shown in Figure 6b. Before post-processing, there is a marked difference of the empirical climacogram from the theoretical. This does not indicate a weakness of the algorithm. It just reflects the fact that, with a Hurst parameter as high as $H = 0.92$, there is high uncertainty and variability, while a sample of $n = 10\,000$ is too short to eliminate this uncertainty; note that the equivalent sample size (which indicates the sampling variability) in this case is $n' := \gamma(1)/\gamma(n) \approx 7$ instead of $n = 10\,000$. Interestingly, the post-processing substantially decreases the difference from the theoretical curve. The improvement due to post-processing is spectacular in panel (d), which shows a comparison of the theoretical and empirical marginal distribution of $\underline{x}_\tau$. Before post-processing, even though the cumulants are preserved, the initially generated small values are problematic as no zero values are generated. This is fully remedied by the post-processing technique. Finally, panel (c) shows that the autocorrelations are well preserved both before and after post-processing.

Further information on the form of the generated time series is provided in Figure 7, this time showing not the statistical characteristics, but the time series per se. The plot, covering a period of 2000 h (83 d; panel (a)) with a focus on the first 200 h (~8 d; panel (b)), indicates that the time series resemble the form of natural rainfall events.

### 3.5. Simulating the Precipitation Process at the Annual Time Scale

The same precipitation model as in the previous subsection was used for generation at the annual scale. Now the distribution is no longer Pareto but PBF, whose treatment is more laborious. On the other hand, the probability dry at the annual scale is zero, and thus the distribution is continuous. This makes the generation simpler as no post-processing is required.

While at the hourly scale all cumulants are positive, tending fast to infinity (Figure 6a), at the annual scale, some of the cumulants (most notably the fourth) are negative (Figure 8a). According to the model, again the cumulants tend to infinity, but for much higher $p$ (>33) as now $\xi' = 0.030$. The other parameters of the PBF distribution are $\zeta^{(v)} = 4.00$ and $\lambda^{(v)} = 0.089$ mm/h. The approximation $\underline{v}'_\tau$ of $\underline{v}_\tau$ is made by another PBF distribution with slightly different parameters, $\zeta^{(v')} = 4.01$ and $\lambda^{(v')} = 0.098$ mm/h. As seen in Figure 8a, the achieved approximation is good, except for a substantial difference in the first cumulants of $\underline{v}'_\tau$ and $\underline{v}_\tau$, so that the required shift of Equation (42) is not negligible, $c = 0.0871$ mm/h.

Comparisons of the theoretical statistical characteristics of the distribution of $\underline{x}_\tau$ to the empirical ones of the generated sample are shown in the panels of Figure 8. In all panels, the agreement between theoretical and empirical characteristics is very good.

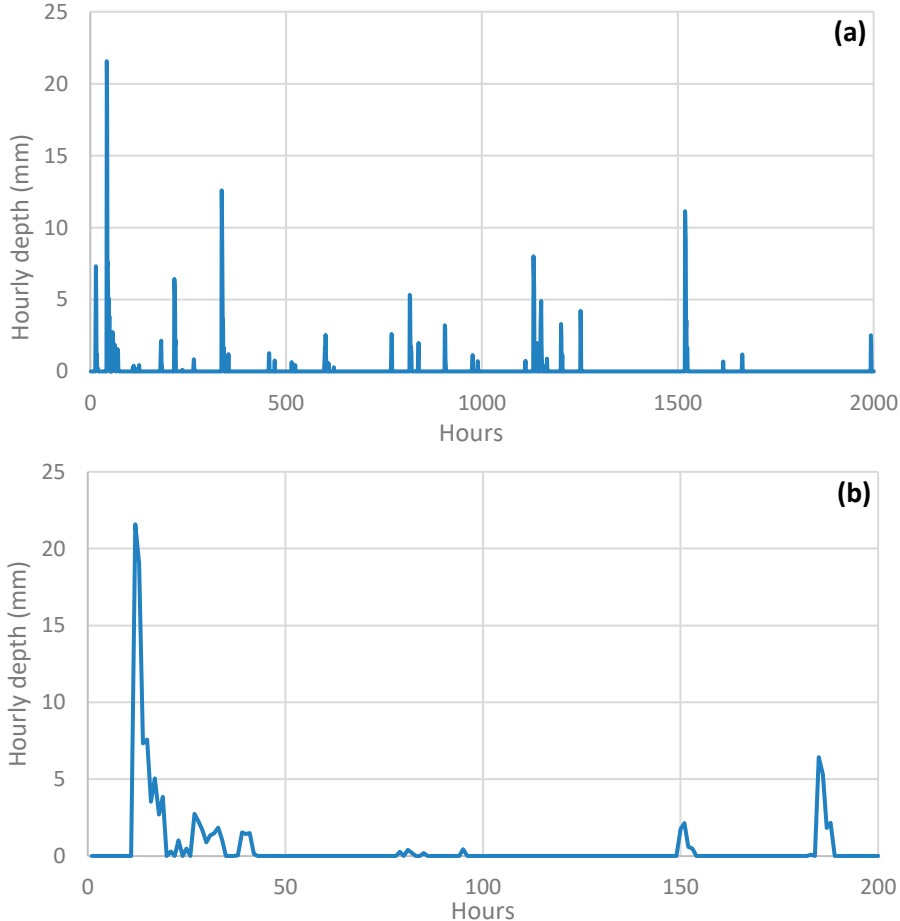

**Figure 7.** Plots of generated time series of precipitation in Bologna at hourly time scale: (**a**) for a period of 2000 h (83 d); (**b**) focus on the first 200 h (~8 d).

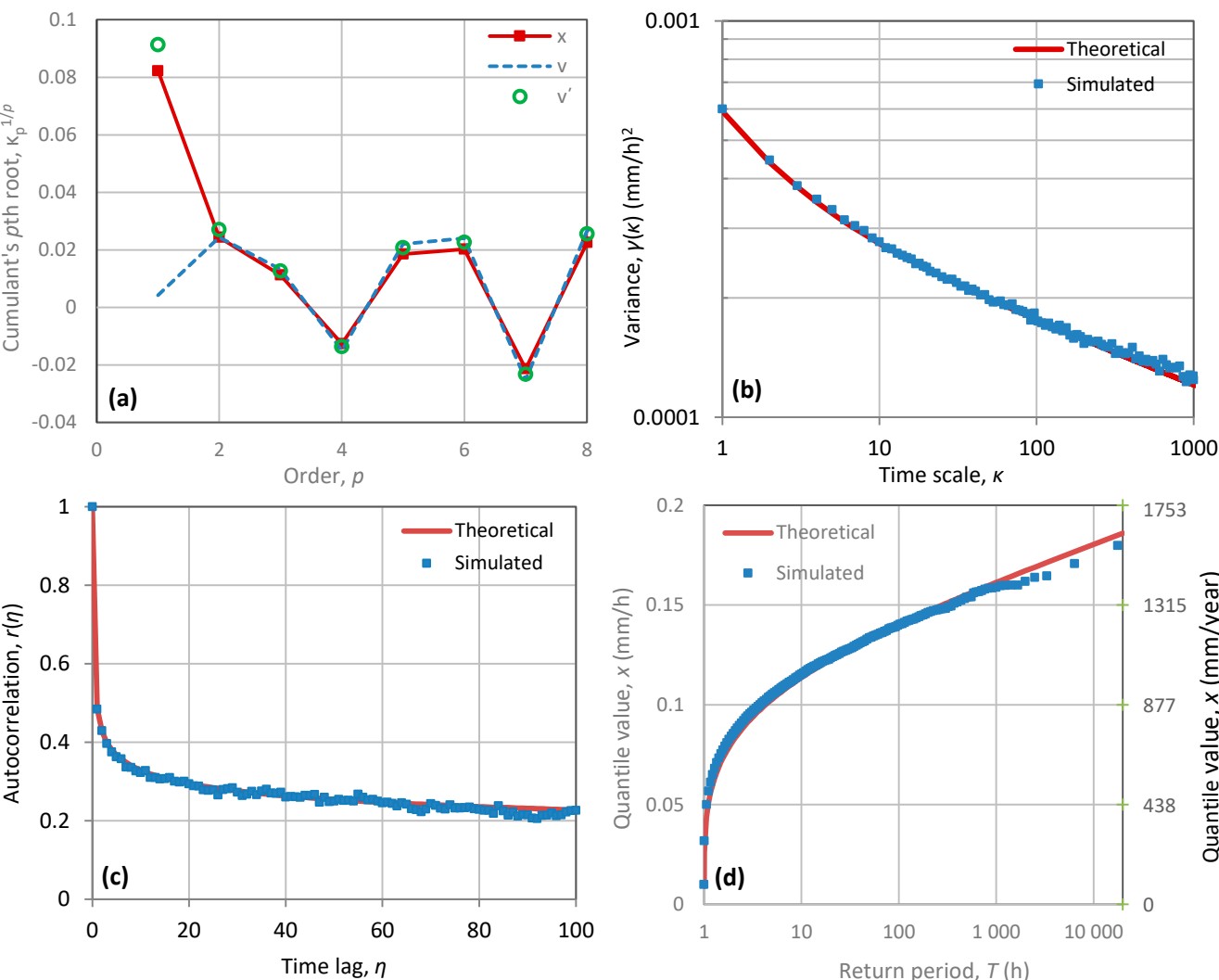

**Figure 8.** Graphical depiction of the results of the simulation application for a real-world case study for the precipitation process in Bologna at the annual time scale, modelled as a persistent FHK process with PBF distribution: (**a**) cumulants; (**b**) climacogram; (**c**) autocorrelogram; (**d**) marginal distribution.

## 4. Discussion and Conclusions

Stochastic simulation of complex processes necessarily relies on approximations of distribution functions. Typically, these approximations are made with reference to the normal distribution, e.g., the Gram–Charlier series, the Edgeworth approximation, etc. [37,51]. These, however, are not good for simulation as no generic random number generation algorithms are available for such type of approximations. They can also be too complicated. Here, we provide more general and more powerful approximations of distribution functions based on cumulants. These are quite flexible and can have several forms, such as (a) the sum of a few (e.g., two or even just one) stochastic variables with typical distributions of an appropriate type (such as those contained in Tables 2–4); (b) the occasional involvement of discontinuities in constituent distributions (usually at their lower bounds); and (c) the discretization of the stochastic variable, in the case that its domain is bounded from both above and below. As random number generation algorithms are readily available for these typical distributions, the proposed approximation is useful in stochastic simulation.

The approximation of a distribution via cumulants turns out to provide very powerful means for stochastic simulation of processes of any type, with short- and long-range dependence. The combination of this approximation with the asymmetric (AMA) or symmetric (SMA) moving average generation schemes can tackle demanding simulation

problems. The genuine stochastic simulation approach that is studied, which does not perform transformations of the stochastic variables involved, is useful, convenient, and powerful. This is particularly the case for problems where time directionality is important; it is reminded that a Gaussian process, even when (back) transformed to non-Gaussian by any nonlinear transformation, cannot provide a process with time asymmetry.

The case studies conducted confirm the excellent performance of the method for a variety of demanding problems and a variety of distributions and time scales. In particular, the long-range dependence, however high, as well as the antipersistence, do not entail any difficulty in applying the method. In contrast, some characteristics of the marginal distribution, such as single or double boundedness, and especially the possible intermittence, may cause difficulties. For this reason, all case studies conducted involve non-Gaussian marginal distributions that are bounded, thus making the problems more challenging. These include distributions double-bounded, such as uniform, and single-bounded, such as exponential, Pareto and PBF, with the Pareto distribution also having a discontinuity at the origin (intermittence). The examples studied show how the problems of boundedness and discontinuity can be handled through simple post-processing procedures, thus achieving an overall satisfactory performance.

In conclusion, the method seems promising and expandable to several future research directions, such as multivariate stochastic modelling, downscaling, disaggregation, and stochastic modelling of two or more processes simultaneously, particularly in cases where time directionality is important (e.g., rainfall-runoff modelling at small time scales).

Stochastic simulation has recently acquired tremendous importance, as conventional energy sources are being replaced with renewables, whose nature is stochastic and, thus, their assessment needs stochastic tools. Its utility should now be appreciated more than ever, after various spectacular failures of aspirations to achieve satisfactory predictions of geophysical processes in deterministic terms, and after reconciliation with the fact that uncertainty is an intrinsic characteristic of nature, not subject to elimination.

**Author Contributions:** Conceptualization, D.K. and P.D.; methodology, D.K.; software, D.K. validation, D.K. and P.D.; formal analysis, D.K. and P.D.; investigation, D.K. and P.D.; resources, D.K.; data curation, D.K.; writing—original draft preparation, D.K.; writing—review and editing, D.K. and P.D.; visualization, D.K.; supervision, D.K.; literature review, D.K. and P.D. All authors have read and agreed to the published version of the manuscript.

**Funding:** This research received no external funding; it was conducted for scientific curiosity.

**Institutional Review Board Statement:** Not applicable.

**Informed Consent Statement:** Not applicable.

**Data Availability Statement:** Not applicable.

**Acknowledgments:** Discussions with Theano Iliopoulou, Federico Lombardo, and Ioannis Tsoukalas helped us with the method conceptualization. We thank the two anonymous reviewers for the positive evaluation of the paper and their helpful comments.

**Conflicts of Interest:** The authors declare no conflict of interest.

## Appendix A  Comparison with a Conventional Approach

This Appendix A (not contained in version 1 of our paper) was added, following a suggestion by an anonymous reviewer that it would be good if the paper contained comparisons with traditional approaches, which include transformations from Gaussian processes. As a traditional approach we choose the ARMA(1,1) model, and as a case study, we use the one presented in Section 3.1, which deals with the exponential distribution. (The case studies of the Sections 3.2–3.5 can hardly be dealt with using traditional approaches). As the exponential distribution assumed in this case study is a special case of the gamma distribution, we use the traditional Wilson–Hilferty–Kirby transformation [52], which transforms a standard Gaussian variable $\underline{z}$ to a variable $\underline{w}$ with approximately (three-

parameter) gamma distribution with mean 0, standard deviation 1 and coefficient of skewness $C_s$. In its original (Wilson–Hilferty) form, the transformation is:

$$\underline{w} = \frac{2}{C_s} \left( \left( 1 - \left( \frac{C_s}{6} \right)^2 + \frac{C_s}{6} \underline{z} \right)^3 - 1 \right) \tag{A1}$$

Kirby [52] gave a better approximation by modifying the transformation in the following form:

$$\underline{w}^M = A \left( \max \left( C, 1 - \left( \frac{D}{6} \right)^2 + \frac{D}{6} \underline{z} \right)^3 - B \right) \tag{A2}$$

where $A, B, C, D$ are coefficients depending on $C_s$, given by Kirby [52] in tabulated form, except for $C$, which is calculated as

$$C = \left( B - \frac{2}{C_s} \frac{1}{A} \right)^{\frac{1}{3}} \tag{A3}$$

Plugging $C$ in Equation (A2) we see that if the value of $z$ is too low (strongly negative), then the lowest admissive value of $w^M$ is $-2/C_s$. For the exponential distribution, $C_s = 2$ and the tabulated values are $A = 1.03571, B = 0.99968, D = 1.93606$, while $C$ is calculated to 0.32446. We note that the so-calculated variable $\underline{w}^M$ has lower bound $-1$, and hence to achieve the standard exponential distribution we have to take $\underline{w}^M + 1$.

The ARMA(1,1) model for the Gaussian process $\underline{z}_\tau$ is

$$\underline{z}_\tau = a \underline{z}_{\tau-1} + \underline{v}_\tau + b \underline{v}_{\tau-1} \tag{A4}$$

where $\underline{v}_\tau$ is Gaussian white noise with mean 0 and variance $\sigma_v^2$, and $a$ and $b$ are model parameters. Given the model parameters, the autocovariance $c_\eta$ of the process is given as follows [2]:

$$c_0 = \left( 1 + \frac{(a+b)^2}{1-a^2} \right) \sigma_v^2, \ c_1 = a c_0 + b \sigma_v^2, \ c_\eta = a^{\eta-1} c_1, \ \eta \geq 1 \tag{A5}$$

In our case study, we have $c_0 = 1$, $c_1 = 0.701$, $c_2 = 0.509$, while the model cannot preserve autocovariances for lag higher than 2. The resulting model parameters (obtained by a solver) are $\sigma_v^2 = 0.509$, $a = 0.727$, $b = -0.0517$.

We expect that the approximate transformation (A2), by construction, will give variance 1, which in the case study is equal to $c_0$. However, there is no guarantee that the values of $c_1$, $c_2$ will be preserved after applying the transformation. An analytical calculation of the values of $c_1$, $c_2$ after the transformation is not possible and therefore, we have to resort to numerical methods [19–24], of which a Monte Carlo method is the easiest. However, for simplicity here we assume that the changes in the autocorrelations, $\rho_1 = c_1/c_0$, $\rho_2 = c_2/c_0$ are negligible. With this assumption, we easily run the model to generate 10 000 synthetic values, from which we constructed Figure A1. This should be viewed in comparison to Figure 1. One can see in Figure A1c that the transformation is satisfactory in preserving the marginal distribution. The problems appear in the climacogram and the autocorrelogram. Clearly, the conventional ARMA model cannot reproduce the LRD. On the other hand, the autocorrelations $\rho_1$, $\rho_2$ are preserved and indeed the changes due to the transformation are negligible, which confirms the validity of our assumption. We note though that there exist more sophisticated methods, relying on transformations to Gaussian, which can preserve the LRD (e.g., [24]), but these do not classify as conventional approaches. Yet the method proposed here, which is quite generic and preserves high order moments in a genuine manner, enables potential application in even more demanding cases, such as when the time's arrow is important to handle, as already mentioned in the introduction.

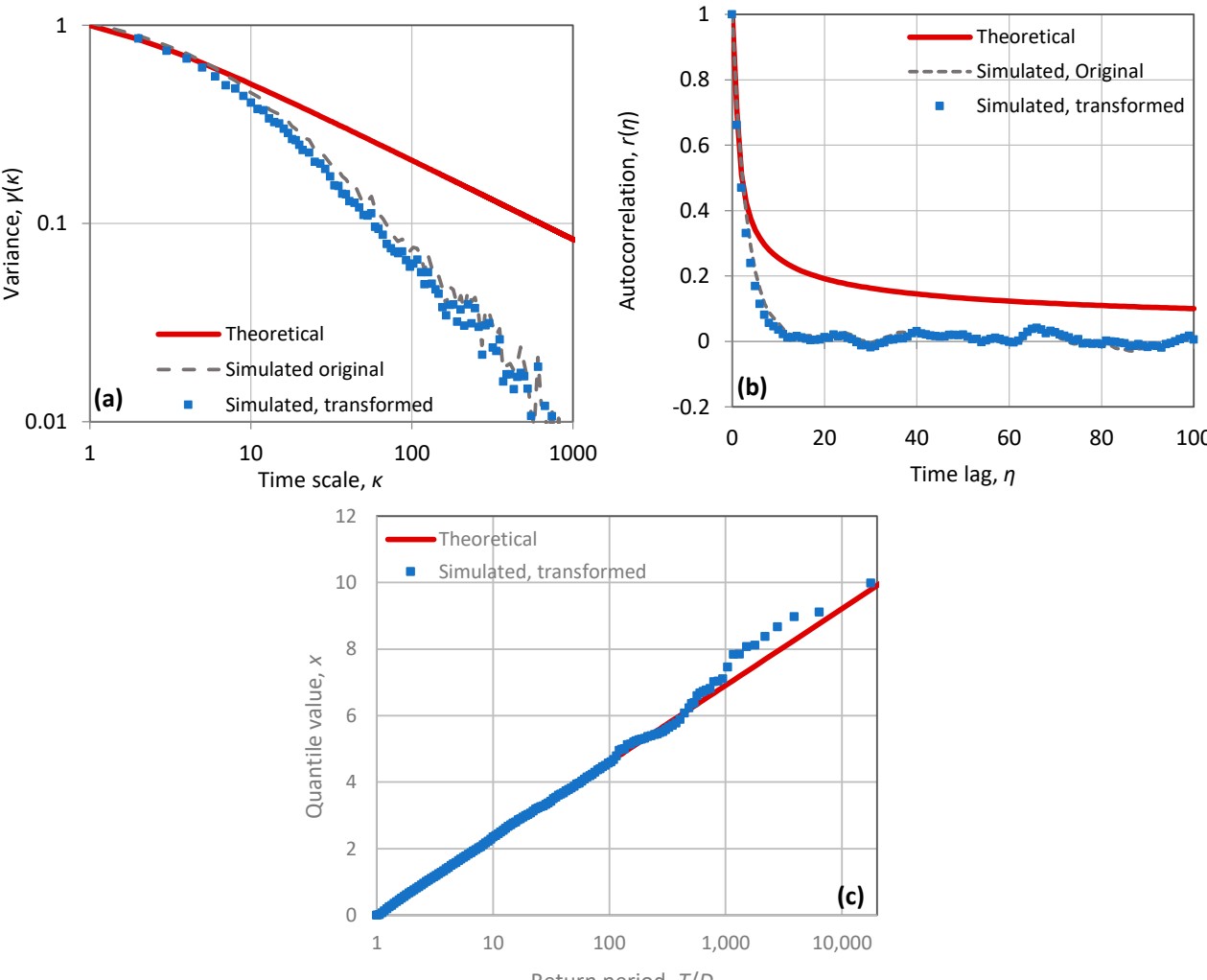

**Figure A1.** Graphical depiction of the results of the simulation application for a synthetic example of an ARMA(1,1) model as an approximation of the FHK process in the case study of Section 3.1, with an exponential distribution: (**a**) climacogram; (**b**) autocorrelogram; (**c**) marginal distribution. The figure should be viewed in comparison to Figure 1 (panels b–d, respectively).

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
