# Peer review of "Towards Generic Simulation for Demanding Stochastic Processes"

_sci, doi:10.3390/sci3030034_

Round 1

Reviewer 1 Report

This paper proposed a new genuine simulation method for stochastic processes, which can generate non-Gaussian processes. It provides very important findings to the stochastic processes that exists almost in every geophysical process that would appeal to readers in this field. The manuscript is well organized and written that is almost ready to be published.

The developed method provides a general approximation to any type of stochastic processes and will be widely applicable. From the case studies, it is easy to see the powerfulness and effectiveness of the new method, however, it would be better if comparisons from the result of traditional approaches that need to transfer to Gaussian processes, can be given so that the advantages of the proposed method can be directly seen from figures.

In the introduction, it might be difficult to follow if the reader does not have any idea of time series analysis or mathematics, therefore, some information in plain language would be recommended.

Author Response

We are grateful to the reviewer for the positive evaluation and the helpful comments.

In response to the comment "it would be better if comparisons from the result of traditional approaches that need to transfer to Gaussian processes, can be given so that the advantages of the proposed method can be directly seen from figures" we have now added the Appendix entitled "A comparison with a conventional approach".

In response to the comment "In the introduction, [...] some information in plain language would be recommended" we have added the following text in the Introduction:

The new methodology advances the state-of-the-art in stochastic generation by providing a general framework capable of dealing with challenging Monte-Carlo applications within geophysics, engineering and other fields. The merits of the methodology rely on its ability to cope with the following aspects:

  1. Complex dependence structures that extend way beyond the Markov dependence, and incorporate long-range dependence and short-scale fractal (smoothness/roughness) behaviour. This is achieved by using a symmetric moving average scheme, which can involve a large number of white noise terms, with their weights determined in an explicit analytical manner.
  2. Marginal distributions that extend beyond Gaussian and incorporate heavy tails, boundedness and intermittence. This is achieved by using an appropriate number of cumulants, analytically determined from the distribution function, thus resulting in genuine simulation of the process (without a transformation).
  3. Time asymmetry (irreversibility), achieved by using a non-Gaussian distribution function, combined with an asymmetric moving average scheme, with the weights again determined in an explicit analytical manner.

Reviewer 2 Report

The authors have presented a method to obtain approximations of distribution functions on the basis of cumulants for use in stochastic simulations. Several applications with persistent and antipersistent processes are shown with comparison between theoretical and simulation results. The authors must be commended to have provided a very precise review of previous work and theoretically sound approach.

Author Response

We are grateful to the reviewer for the positive evaluation and the flattering comments.